# Targeted Treatment Options of Recurrent Radioactive Iodine Refractory Hürthle Cell Cancer

**DOI:** 10.3390/cancers11081185

**Published:** 2019-08-15

**Authors:** Mehtap Derya Aydemirli, Willem Corver, Ruben Beuk, Paul Roepman, Nienke Solleveld-Westerink, Tom van Wezel, Ellen Kapiteijn, Hans Morreau

**Affiliations:** 1Department of Medical Oncolosgy, Leiden University Medical Center, 2333 ZA Leiden, The Netherlands; 2Department of Pathology, Leiden University Medical Center, 2333 ZA Leiden, The Netherlands; 3Hartwig Medical Foundation, 1098 XH Amsterdam, The Netherlands

**Keywords:** Hürthle cell carcinoma, thyroid neoplasm, near-homozygous genome, targeted therapy, panitumumab, humans, cell line, tumor, cell signaling, epidermal growth factor receptor, AKT

## Abstract

*Objective*: To evaluate the efficacy and treatment rationale of Hürthle cell carcinoma (HCC) following a patient with progressive and metastatic HCC. HCC was recently shown to harbor a distinct genetic make-up and the mitogen-activated protein kinase (MAPK) and phosphatidylinositol 3-kiase (PI3K)/AKT signaling pathways are potential targets for anti-cancer agents in the management of recurrent HCC. The presence or absence of gene variants can give a rationale for targeted therapies that could be made available in the context of drug repurposing trials. *Methods:* Treatment included everolimus, sorafenib, nintedanib, lenvatinib, and panitumumab. Whole genome sequencing (WGS) of metastatic tumor material obtained before administration of the last drug, was performed. We subsequently evaluated the rationale and efficacy of panitumumab in thyroid cancer and control cell lines after epidermal growth factor (EGF) stimulation and treatment with panitumumab using immunofluorescent Western blot analysis. EGF receptor (EGFR) quantification was performed using flow cytometry. *Results:* WGS revealed a near-homozygous genome (NHG) and a somatic homozygous *TSC1* variant, that was absent in the primary tumor. In the absence of *RAS* variants, panitumumab showed no real-life efficacy. This might be explained by high constitutive AKT signaling in the two thyroid cancer cell lines with NHG, with panitumumab only being a potent inhibitor of pEGFR in all cancer cell lines tested. *Conclusions:* In progressive HCC, several treatment options outside or inside clinical trials are available. WGS of metastatic tumors might direct the timing of therapy. Unlike other cancers, the absence of *RAS* variants seems to provide insufficient justification of single-agent panitumumab administration in HCC cases harboring a near-homozygous genome.

## 1. Introduction

Differentiated non-medullary thyroid cancers (DTC), account for the vast majority (~95%) of thyroid cancers. These can be divided into main histologic types papillary (PTC), follicular thyroid carcinoma (FTC) including subtype variants, and Hürthle cell carcinoma (HCC) [1,2]. Usually, prognosis is favorable with standard therapy including total thyroidectomy followed by radioactive iodide (RAI) and thyrotropin suppression therapy, which cures the majority of DTC cases [3]. However, 5% to 15% of cases become recurrent or metastatic, of which 26% to 60% [4,5,6] progress to RAI refractory status. The refractory DTC cases form a major source of thyroid cancer-related deaths with a 10-year survival of less than 10% [5,7]. Remarkably, of the recurrent cases, a substantial share consists of *BRAF* mutated PTC and HCC [8,9,10]. HCC makes up 3% to 7% of all differentiated thyroid cancers [11], but ranges from 10.5% to 43% in recurrent cases [11,12,13,14].

Typically, HCC features include highly eosinophilic cytoplasm due to accumulated dysfunctional mitochondria [15,16] with frequent mtDNA complex I genetic variations [17] and whole-chromosome losses with retention especially of chromosome 7 [18,19,20,21,22]. The latter manifests as a near-homozygous genome (NHG), which results from near-haploidization frequently followed by endoreduplication or whole genome doubling [18,19,20,21,22]. In particular, HCCs with widespread losses of whole chromosomes appear more likely to progress and recur [8,20,23].

Major drivers of thyroid cancer include the mitogen-activated protein kinase (MAPK) and the phosphatidylinositol 3-kiase (PI3K)/AKT signaling pathway [24,25,26,27]. Drivers of PI3K-AKT signaling include G-coupled protein receptors, tyrosine kinase receptors, and mutant *RAS* [28]. *RAS* mutations dually activate the MAPK and PI3K-AKT pathways. In thyroid carcinogenesis, the latter appears to be more preferred as suggested by the association with AKT activation [27,29,30]. Strong PI3K/AKT signaling and constitutive AKT phosphorylation has been reported in HCC and is found to differentiate adenoma from widely invasive HCC, which appears to be a potential mechanism in HCC development [31]. In a recent genetic study of thyroid cancers, the PI3K/AKT signaling pathway was affected in 23% of HCC analyzed [32]. At the same time, several genetic alterations have been described that could lead to increased pAKT [20,21,27,33]. Furthermore, enrichment of genes involved in PI3K/AKT/mTOR and RAS/RAF/MAPK pathways may ensue from endoreduplication/whole chromosome doubling of chromosomes 5, 7, and 12 that are reported to contain these genes [20,33].

In the course of disease, systemic or molecular-targeted therapies may be considered [34]. For example, therapies targeting mitogen-activated protein kinase (MAPK, or ERK) prolonged progression-free survival in thyroid cancer [34]. Nonetheless, effective treatments that improve overall survival compared to the placebo is lacking. American Thyroid Association (ATA) guideline recommendations for management of recurrent and metastatic disease include the consideration for clinical trials [3,35]. Recent genomic studies revealed new DNA variants in HCC [20,21,32]. Unfortunately, specific recommendations for management of HCC do not exist [36]. In this study, the clinical course and state-of-the-art treatment options, including participation in (basket) trials, in an HCC patient is illustrated.

In the Netherlands, the Drug Rediscovery Protocol (DRUP) basket trial (NCT02925234) was initiated for selecting patients to be administered off-label targeted therapies based on their tumoral mutational profiling, rather than the regular indication range concerning the cancer type it is originally registered for. As such, panitumumab and cetuximab are part of the standard care for advanced colorectal cancer (CRC) [37,38]. These EGFR inhibiting monoclonal antibodies are administered in case of wildtype *KRAS* or *NRAS* genes and target the extracellular part of EGFR [39]. For that reason, the option of EGFR inhibitors in *RAS* wildtype non-CRCs is part of the DRUP trial. In this study, we examined the clinical effects of several targeted drugs in a patient with HCC for which whole genome sequencing of metastatic tumor material was performed at an advanced stage of his disease. With HCC reportedly having lower prevalence rates of *RAS* mutation than FTC [31,32,40,41], the present HCC patient had wildtype *RAS* and was considered for panitumumab therapy. Considering the background of HCC, the impact of panitumumab on its receptor of target and downstream major signaling activators, AKT and ERK, involved in thyroid tumorigenesis, was analyzed in an HCC cell line model with a NHG.

## 2. Results

### 2.1. Treatment Options of Recurrent HCC

Possible treatment options for recurrent HCC are illustrated by a 40-year-old male patient with a left sided widely invasive HCC (5.3 cm in diameter) with extensive vaso-invasion. No lymph node metastasis was found (0/17). The patient had no known risk factors, no intoxications, and his family history included no cancers except for one third-degree family member who was curatively treated for thyroid cancer at the age of 39. Surgical resection of the thyroid with lymph node dissection, radio-iodide therapy, and TSH suppression was performed five years prior to his death. On pre-operative and post-operative I-131 therapy scan, iodide accumulation was detected in the bone, which is indicative of metastases. Five months after this radio-iodide therapy, Positron emission tomography-computed tomography (PET-CT) showed vitality of these bone metastases with rising thyroglobulin, which comply with radio-iodide refractory disease. For the metastatic osseous lesion, the patient received radio-frequent ablation and cementation, which was followed by radiotherapy. Six months later, a PET-CT scan showed residual/recurrent metastasis at the same osseous site with additional novel metastatic lesions in the vertebral column (C7) and liver.

The patient was consecutively treated with various tyrosine kinase inhibitors (see Materials and Methods section) in or out of clinical trials with consideration of clinical status along with assessment of overall lesion response (Figure 1). At this stage, no molecular testing was done.

Upon giving everolimus, there was a stable disease at 12 weeks. Adverse effects included mild fatigue, rash, acne, moderate mucositis and were manageable by using local triamcinolone, lidocaine, and daktarin gels. However, progressive disease was observed at 25 weeks, which initiated a switch to sorafenib. After one week, the patient developed signs of intolerable toxicity with severe palmo-plantar dysesthesia syndrome with severe blistering, legitimizing discontinuation followed by restart at a reduced dose, which reiterates toxicity and ditto implications. On nintedanib, a seven-week course of stable disease was followed by progressive disease at 16 weeks. Adverse events included mild fatigue and alopecia. The patient also received radiation on a vertebral metastatic lesion and stereotactic radiotherapy on a new extra-axial dural lesion that developed before the initiation of nintedanib. On lenvatinib, a partial response was seen over one year of treatment, which was his best overall response (BOR). From lenvatinib use, hypertension ensued initially without need for antihypertensive drugs. Other adverse events included mild diarrhea, fatigue, and raised TSH levels (acted upon by adjustments of the levothyroxine dose). Five months further, progressive disease was determined. However, 10 months of continued use was added out of clinical benefit. Bone metastases had become more expansive and liver metastases were progressive as well, whereupon lenvatinib was ceased eventually. On the vertebral lesion (C7), embolization was done, when the risk of inflicting harm to the myeloid would be too high for radiotherapy.

In the context of the DRUP trial, whole genome sequencing (WGS) was performed on biopsy material from a liver metastasis, which showed loss of heterozygosity (LOH) of whole chromosomes on all chromosomes except for chromosome 5, 7, and 20 (Figure 2).

This NHG phenotype is characteristic for HCC [18,19,20,21]. In addition, homozygous somatic variants were found in *TSC1* (c.611G > T, p.Arg204Leu, VAF 74%, class 3–4) and in *NF1* (c.2750T > G, p.Val917Gly, VAF 82%, class 3). Other detected homozygous somatic variants included *TP53* (c.607G > T p.Val203Leu, VAF 85%, class 3–4) and *RB1* (c.2239G > T p.Glu747 *, VAF 85%, class 4–5) During the molecular tumor board meeting both the *TSC1* and *NF1* variants were considered a potential candidate target rendering sensitivity to mTOR inhibition (Figure 3), even though the initial treatment with everolimus had not, evidently, been successful in this patient.

We subsequently sequenced the primary tumor, which harbored the same homozygous *NF1* variant (VAF 82%). As opposed to the metastatic lesion, next-generation sequencing (NGS) of the primary tumor did not reveal the *TSC1* variant. In addition, in contrast to the metastatic lesion after a five-year disease interval, the primary tumor appeared heterozygous for chromosome 12 based on a SNP in *CDKN1B* (c.326T > G, p. (Val109Gly)), whereas the metastatic hepatic lesion revealed homozygosity.

The absence of *RAS* variants motivated the prescription of panitumumab, which was made available within the DRUP basket trial. Two cycles of panitumumab were administered, but no clinical improvement was noted. The patient was quickly progressive during the panitumumab regimen and his clinical state deteriorated on a pathologic vertebral fracture.

### 2.2. Rationale for Non-Responsiveness to Panitumumab

We further addressed the occurrence of non-responsiveness in the HCC index patient toward the EGFR receptor directed inhibition therapy using panitumumab in thyroid cancer cell line models for HCC or FTC with an NHG. These are cell lines XTC.UC1, FTC-236 [23]. XTC.UC1 has not been fully characterized in the literature. Using NGS we found, apart from NHG, a homozygous *TP53* variant (c.451C > A, NP_000537.3: p. (Pro151Thr)), the mutational load was low (11.02 Mutations/Mb) and no other pathogenic variants of gene fusions were found. These were compared to thyroid cancer cell lines SW579, BHP 2-7, and TT2609-C02.

#### 2.2.1. EGFR Quantification by Flow Cytometry

EGFR was quantified using flow cytometry using A431 and C33A cells as a positive and negative control, respectively. The TC cell lines all expressed the EGF-receptor on their cell membranes and could be subdivided into two groups: one group (SW579 and XTC.UC1) with values ranging from approximately 120,000 to 135,000 MEF and a second group (BHP 2-7, FTC-236, TT2609-C02) with values ranging from approximately 30,000 to 50,000 MEF. A plateau phase was reached at approximately 2 μg panitumumab (Figure 4).

#### 2.2.2. Immunofluorescent Western Blotting Analysis

Comparable to flow cytometry, Western blotting showed EGFR overexpression in A431 cells (positive control) as compared to all thyroid cell lines, with XTC.UC1 and SW579 showing relatively more EGFR compared to BHP 2-7, FTC-236, and TT2609-C02 (Figure 5). Prior to analyzing the impact of panitumumab on cell signaling, we performed cell proliferation analysis and flow cytometric analysis of cell cycle distribution and found no substantial effects (see Appendix A). For Western blotting analysis, cells were left untreated, treated with panitumumab, EGF and panitumumab, or EGF alone. EGF clearly induced autophosphorylation of EGFR at Tyr1148 in four out of the five TC cell lines compared to the control (Figure 5), except for TT2609-C02, which harbors a *NRAS* mutation. Panitumumab significantly reduced the levels of EGFR phosphorylation to control levels. Panitumumab alone did not alter the phosphorylation status of the EGF receptor (control). EGF induced ERK phosphorylation in FTC-236, XTC.UC1, SW579, and in the control cell line A431. However, BHP 2–7 (with a *RET*/PTC1 rearrangement) and TT2609-C02 (harboring a *NRAS* mutation) seem to remain unaffected by EGF or panitumumab regarding ERK phosphorylation. These cell lines show constitutive ERK phosphorylation irrespective of the treatment. Panitumumab could only partly inhibit EGF-induced ERK phosphorylation. Additionally, AKT is constitutively active in these cell lines, at low levels in TT2609-C02. EGF induced additional AKT phosphorylation in XTC.UC1 and in A431. Of interest, AKT phosphorylation was significantly higher in TC cell lines FTC-236 and XTC.UC1, that harbor a near-homozygous genome (NHG) [23], compared to BHP 2–7, SW579, and TT2609-C02 (non-NHG).

## 3. Discussion

In the present study, we illustrate the treatment options for patients with recurrent and metastatic HCC of the thyroid (with NHG) following the targeted treatments chosen for a single HCC patient with metastatic disease.

Various targeted agents (everolimus, sorafenib, nintedanib, lenvatinib, and panitumumab) were administered, based on approved status, or availability within the frame of trials, consecutively, in a time frame of four years. Because of a quick clinical progression due to metastasis, the patient’s death was within five years after surgery.

Everolimus has been studied in thyroid cancer patients in phase II trials showing a median PFS of 11.8 months [42] (similar to the outcome in our patient) and nine months [43], respectively. A combination with other agents, e.g., lenvatinib or sorafenib, is currently being studied (clinicaltrials.gov).

For sorafenib, results of the phase III DECISION trial led to FDA approval, with a median PFS of 10.8 months versus 5.8 months in the placebo arm [10]. The current patient experienced intolerable toxicity due to this drug leading to discontinuation. While, on the other drugs, adverse events were rather well-managed with conservative measures. Among other patient specific factors, the patient’s young age and low WHO performance status may have contributed to the tolerability for the agents and may not necessarily be representative for the real-life target population. Intolerability for adverse events of these targeted agents is very common and may range in seriousness from necessitating additional conservative or invasive measures, according to the grade, dose reductions, temporary withdrawal, or discontinuation [34,44].

Nintedanib, a triple angiogenesis inhibitor, was shown to slightly prolong median PFS to 3.71 months in advanced DTC, compared to 2.86 months in the placebo arm, in a recent phase II trial [45]. In the present patient, SD was denoted at approximately two months of nintedanib use, before progressive disease at four months.

In retrospect, we saw a long-lasting response after giving lenvatinib, with partial response (PR) at 14.25 months as the best overall response. This multi-kinase inhibitor is one of the approved drugs for the treatment of progressive, refractory differentiated thyroid cancer. In the randomized phase III SELECT trial, lenvatinib has been shown to increase progression-free survival (PFS) with 14.7 months compared to a placebo, a 64.8% response rate, and an 87.7% disease control rate [7]. A recent update showed a median PFS of 19.4 months versus 3.7 months for the placebo. Among lenvatinib-treated patients, responders (defined as complete response (CR) or PR as the best overall response according to RECIST) had a PFS of 33.1 months versus 7.9 months for non-responders [46].

When exploring potential further directions for personalized therapy based on molecular profiling in the current patient, whole genome sequencing was performed on metastatic biopsy material. HCC carrying a near-homozygous genome was revealed, with loss of heterozygosity on all autosomes except for chromosomes 5, 7, and 20. Tumors with extensive NHG, which is a characteristic HCC feature [18,19,20,21], appeared to correlate with gene overexpression in the mTOR pathway and protein translation [20].

Furthermore, a potential target for mTOR inhibitory treatment, the homozygous *TSC1* mutation, was detected upon WGS analysis of metastatic material. Various cancer types have been shown to carry druggable targets for mTOR inhibitors, including mutations in *MTOR*, *TSC1*, *TSC2*, *NF1*, *PI3KCA*, *PIK3CG*, *STK11*, and *RHEB* [47,48,49,50,51,52,53,54]. Among these, an ATC patient was reported by Lim et al. with a *TSC1* nonsense mutation, showing clinical benefit to everolimus [54]. Wagle et al. reported the case of a patient with an ATC, that was derived from an HCC carrying a homozygous *TSC2* nonsense mutation rendering striking sensitivity to everolimus with an 18-month response. Progressive disease followed due to an acquired mutation in *MTOR*, which confers resistance to allosteric mTOR inhibition [52].

Furthermore, in the metastatic material from the present patient, chromosome 12 revealed LOH, as opposed to the primary resected lesion five years prior. Presumably, this is indicative of the gradual progression of whole chromosome losses, as seen in this type of tumors. With chromosomes 5, 12, 20, and especially chromosome 7 often being exempt from LOH [18,19,20,21], the homozygous chromosome 12 could be recognized as a relatively advanced-stage event in this tumor site. In addition, the *TSC1* homozygous pathogenic variant (on chromosome 9) detected on metastatic material, was not present in the primary tumor. This would serve as a possible explanation for the patient’s relative unresponsiveness to everolimus earlier in the disease course. In retrospect, the revelation of the newly acquired homozygous *TSC1* variant in the hepatic metastasis could have justified a second attempt for an everolimus regimen. However, the patient’s condition did deteriorate too rapidly for such a choice and everolimus was not available within the DRUP protocol.

Confirmation of wild-type *RAS* status prompted the initiation of panitumumab. A fully humanized IgG2 monoclonal antibody with high affinity and specificity for EGFR, which is part of standard treatment for metastatic CRC [37,38]. However, in the index patient, notwithstanding the limited extent of use ensuing from an infaust disease course, no signs of benefit were achieved with this agent.

The outcome of the cell line experiments may serve as a rationale against anti-EGFR therapy with panitumumab in HCC patients. The non-effectiveness of anti-EGFR receptor treatment might be caused by constitutive AKT signaling especially in XTC.UC1 and FTC-236, that serve as a model for HCC with NHG. Panitumumab, however, appeared to lower ligand stimulated expression of activated ERK, but not to basal levels, which is suggestive of the presence of orphan receptor mechanisms [55].

To the best of our knowledge, panitumumab has not been tested before in HCC and FTC cell line models harboring a near-homozygous genome in particular. Indeed, strong PI3K/AKT signaling and phosphorylated AKT (pAKT) overexpression has been reported in HCC and, moreover, found to differentiate widely invasive HCC from adenoma, which appears to be a potential mechanism in HCC development [31]. Thus, even in the absence of the *RAS* mutation, these cases could be considered unlikely to achieve a potential benefit from panitumumab treatment. The study by Ganly et al. also reports AKT/mTOR activation in HCC [20]. Tumors with ‘whole chromosome doubling’ (or endoreduplication [22]) of chromosome 5, 7, and 12 are reported to contain genes involved in the PI3K/AKT/mTOR and the RAS/RAF/MAPK pathway, which leads to amplification of these genes [20,33]. Several genetic alterations have been described that could lead to increased pAKT, including variants of *PTEN*, *PIK3CA*, *AKT1*, *NF1*, and *RAS* [20,21,27,33]. For instance, hypothetically, as upregulation of AKT can result from decreased *PTEN* activity [56], an NHG with LOH of chromosome 10, which *PTEN* is located on, may also lead to a decrease in *PTEN* activity.

Although the case described involves only one patient, it is rather illustrative of the management course using targeted therapies, considerations, and dilemmas encountered in the application thereof. Options provided by drug repurposing for developing a tailored treatment plan based on molecular genetic analysis, is demonstrated. Simultaneously, the elucidation of insight into the molecular characteristics in cancer types may serve to be crucial in this process as well in order to define the ultimate range of indications. The patient succumbed to his disease at a relatively young age, within five years of a refractory HCC, which is also an exemplification of the poor prognosis of refractory HCC and emphasizes the pressing need for effective therapy.

## 4. Materials and Methods

### 4.1. Patient

The patient presented was enrolled under an earlier study protocol for the administration of everolimus as part of the THYRRAD study approved by the Institutional Review Board of the Leiden University Medical Center (LUMC P10-053, CRAD001CNL08T, ethical approval 14 April 2010) and registered at ClinicalTrials.gov (NCT01118065, Clinical Trials registration 6 May 2010) and for nintedanib (ethical approval local committee LUMC 10 July 2014, 1209-EnTF; NCT01788982, Clinical Trials registration 11 February 2013). Written informed consent was provided by the patient. Formalin fixed paraffin embedded (FFPE) primary thyroid carcinoma samples of the primary study were available. Sorafenib was used as standard therapy and lenvatinib was provided by a Named Patient Program (NPP). Panitumumab was made available to the patient as part of the DRUP trial (ethical approval central committee Dutch Cancer Institute 19 April 2016, NL54757.031.15; NCT02925234, Clinical Trials registration 5 October 2016). Assessment of overall lesion response was done using Response Evaluation Criteria in Solid Tumors (RECIST) 1.1 [57]. Targets of the administered drugs include: Everolimus, mTOR. Sorafenib, BRAF, RAF, KIT, FLT3, VEGFR2-3, and PDGFRβ. Nintedanib, PDGFRα-β, FGFR1-3, VEGFR1-3, FLT3, Lck kinase, Lyn kinase, and Src kinase. Lenvatinib, VEGFR1-3, FGFR1-4, PDGFRα, KIT, and RET. Panitumumab, EGFR [58].

### 4.2. Somatic Gene Variant Spectrum Screening

Tissue biopsy material from the liver metastasis was analyzed using whole genome sequencing (WGS) (Illumina X10 setup, https://emea.illumina.com/company.html#) at the Hartwig Medical Foundation. For WGS analysis, a fresh-frozen biopsy was taken from a metastatic liver lesion following lenvatinib treatment, together with a control blood sample. WGS DNA sequencing was performed and analyzed following standardized procedures described by Priestley et al. [59].

Samples from the primary thyroid resection specimen and from the metastatic material were analyzed using Sanger sequencing for hotspot DNA variants in *KRAS*, *BRAF*, *EGFR*, *PIK3CA*, and tested with a dedicated NGS panel targeting *NF1*, *TSC1*, *CDKN1A*, *CDKN2A*, *CDKN2B*, *CDKN2C*, and *CDKN2D*, as described previously [43].

Isolated DNA from the XTC.UC1 cancer cell line was analyzed using NGS with the Oncomine^TM^ Tumor Mutation Load (TML) Assay and estimation of Tumor mutational burden (TMB) (thermofisher.com/tmb). Isolated RNA from the XTC.UC1 cancer cell line was analyzed using NGS with the Archer FusionPlex CTL Panel for the detection of fusions in target genes *ALK*, *AXL*, *BRAF*, *CCDN1*, *FGFR1-3*, *MET*, *NRG1*, *NTRK1-3*, *PPARG*, *RAF1*, *RET*, *ROS1*, and *THADA* (https://archerdx.com/fusionplex-assays/ctl-rna).

Pathogenicity of DNA variants is annotated in classes, ranging from class 1 to 5 [60]. Class 5: pathogenic variation. Class 4: likely pathogenic variation. Class 3 (other variations): variance of unknown pathogenicity based on literature (Pubmed) search and common or locus specific databases (Mycancergenome, Alamut Visual, NCBI dbSNP, NCBI ClinVar, COSMIC, Jackson laboratory database, LOVD, MD Anderson database), class 2 (unlikely pathogenic), and class 1 (not pathogenic) DNA variations are not reported.

### 4.3. Cell Lines, Cell Culture, DNA Isolation, and CELL Count

The following human differentiated thyroid cancer cell lines were used for assessing panitumumab treatment on EGFR and downstream signaling using Western blot analysis: BHP 2–7, SW579, XTC.UC1, FTC-236, and TT2609-C02 (Table 1).

As a type of control, the epidermoid carcinoma cell line with known EGFR overexpression, A431 cells, was used in this study.

For assessing EGFR quantification using flowcytometry, an additional cell line with known lack of EGFR expression, C33A, was used as a control [71]. Cell lines were authenticated by Short tandem repeat (STR) profiling (GenePrint^®^ 10 system, Promega, Leiden, The Netherlands).

The cell lines were cultured under standard conditions in humidified atmosphere (5% CO_2_, 95% air, 37 °C). RPMI medium 1640 (Cat. No 52400025, Gibco, Life Technologies, Paisley, UK) was used for SW579 cells and TT2609-C02 cells. The other cell lines were maintained in DMEM/F-12 medium (Cat. No 11330032, Gibco, Life Technologies, Paisley, UK). Culture media were supplemented with 10% heath-inactivated Fetal Bovine Serum (Cat. No 758093, Greiner bio-one, Longwood, FL, USA), penicillin (50 U/mL), streptomycin (50 µg/mL) (Cat. No 15140122, Gibco, Life Technologies, Paisley, UK).

Cells were thoroughly washed with Hank’s balanced salt solution (HBSS, Sigma-Aldrich, St. Louis, MO, USA) and harvested using HBSS containing 0.125% trypsin (Gibco, Life Technologies) and 0.5 mM EDTA at 37 °C. For DNA isolation, the NucleoSpin purification kit (Macherey-Nagel GmbH & Co. KG, Düren, BRD) was used on cell pellets, according to the manufacturer’s instructions. DNA concentrations were determined using the Picogreen method (Life Technologies).

Cell counts were performed using AO-DAPI (solution 18, Cat. No 9103018, Chemometec, Allerød, Denmark), diluted 1:20. Each cell line was loaded in quadruplicate using the NC-Slide A8 (Chemometec, Allerød, Denmark) and read out by Automated Cell Analyzer NucleoCounter NC-250 (Chemometec, Allerød, Denmark) with NucleoView NC-250 software (Chemometec, Allerød, Denmark).

### 4.4. Compounds, Western Blotting

Cells were cultured until 70% to 80% confluency. Next, cells were serum starved, stimulated by EGF alone for 10 min, or treated with panitumumab (Amgen Inc., Thousand Oaks, CA, USA) for 1 h (20 µg/mL) and stimulated with EGF (Sigma-Aldrich, St. Louis, MO, USA) for 10 min (100 ng/mL), which is followed by washing with ice cold PBS and lysed with Hot-SDS buffer containing PhosSTOP (Cat. No 04906837001, Roche Diagnostics, Mannheim, Germany and cOmplete (Cat. No 11697498001, Roche Diagnostics, Mannheim, Germany).

Protein concentrations were determined with the Bio-Rad DC^TM^ Protein Assay, according to the manufacturer’s instructions (Bio-Rad Laboratories, Inc., Hercules, CA, USA). Next, 10 µg lysates were mixed with 4× Laemmli Sample Buffer (Cat. No 1610747, Bio-Rad Laboratories, Inc.) containing βME, and heated for 5 min at 100 °C and loaded onto a 1.5 mm 10% acrylamide gel in addition to molecular weight markers (92840000 310014776 LI-COR, Lincoln, NE, USA). Electrophoresis was performed at 50 V throughout the gel. For blotting, the Bio-Rad semi-dry Trans-Blot Turbo Transfer System was used (Limit 25V, constant 2.5A, 15 min). Next, blots were washed in 1×TBS, blocked in Odyssey Blocking Buffer (TBS) (Cat. No 92750000, LI-COR, Lincoln, NE) with 1× TBS in ratio 1:1 for 1 h, washed in 1× TBS, and incubated overnight at 4 °C with antibodies in primary antibody diluent containing 1× TBS, 0.1% Tween-20, 5% BSA (Bovine Serum Albumin, Sigma-A9647, Sigma-Aldrich (St. Louis, MO, USA). The next day, blots were washed in 1× TBS and incubated for one hour with the secondary antibodies in 1× TBS/0,1% Tween-20/5% BSA protected from light.

After washing with 1× TBS, the blots were air dried and imaged on high-resolution with the Odyssey Infrared Imaging System (LI-COR, Lincoln, NE), using intensities 1.5 and 3.0 for the fluorescent color channels red (700) and green (800), respectively. Images were analyzed using Image Studio Lite Ver 5.2 software package (LI-COR, Lincoln, NE).

### 4.5. Antibodies

Primary antibodies: anti-αTubulin Mouse, 1:100,000 (clone: DM1A, Cat. No. 14450282, eBioscience, San Diego, CA, USA); EGFR Rabbit mAb, 1:4000 (Cell Signaling #4267; Leiden, the Netherlands). pEGFR (Tyr1148) Rabbit mAb, 1:1000 (Cell Signaling #4404). Akt Mouse mAb, 1:2000 (Cell Signaling #2920). pAkt (Ser473) Rabbit mAb, 1:1000 (Cell Signaling #9277). Erk 1/2 Mouse mAb, 1:1000 (Cell Signaling #4696). pErk 1/2 (Thr202/Tyr204) Rabbit mAb, 1:2000 (Cell Signaling #4370). Secondary antibodies: green-fluorescent Goat anti-Rabbit IRDye 800CW, 1:10,000 (92632211 LI-COR, Lincoln, NE). Red-fluorescent Goat anti-Mouse IRDye 680LT, 1:10,000 (92668020 LI-COR, Lincoln, NE).

### 4.6. EGFR Expression Analysis by Flow Cytometry

Cells (0.5 million/test tube) were spun down (500 g, 5 min, and 4 °C). Fifty µL in PBS/1%BSA (1% PBA) diluted panitumumab (range 1 ng–2 µg) was added to the cell pellets and gently vortexed. Cells were incubated for 60 min on ice. Next, cells were washed twice with PBS and a secondary reagent was added (GaHIgG1-FITC, 1 µg in 50 µL, SouthernBiotech, Birmingham, AL) to the pellet, mixed, and incubated for another 60 min on ice in the dark. Cells were washed twice with PBS. Lastly, 500 µL of 1% PBA was added containing 1 µM of propidium iodide staining dead cells, vortexed, and stored on ice in the dark for 30 min prior to the analysis on an LSRII flow cytometer (BD Biosciences). A blue 488 nm 20 mW laser was used for excitation. A threshold was set on the FSC detector. The FSC-W vs. FCS-H and SSC-W vs. SSC-H were used to gate out doublets during acquisition. For quantitative flow cytometry, Cyto-CalTM calibration beads (Cat No. FC3MV, ThermoFisher, Fremont, CA) were used according to the manufacturer’s instructions, which allows the presentation of EGFR expression in terms of molecular equivalent fluorochrome (MEF). FITC fluorescence or green fluorescence equivalent (beads) was collected using a 530/30 band pass filter. An FSC-A vs. PI-A fluorescence life gate was used for dead cell discrimination and 20,000 single live cell events were collected for each measurement. A data file contained all events, including debris and aggregates. Data were analyzed using WinList 8 (Verity Software House, Topsham, ME).

## 5. Conclusions

In this study, we described current treatment options for recurrent HCC illustrated by a patient and, in addition, assessed the potential value of panitumumab on major signaling pathways involved in thyroid cancer in surrogate cell line models. Based on the results of this study, we conclude that recurrent HCC cases do not seem likely to benefit from panitumumab and the criterion of *RAS* wildtype status for administration seems insufficient for these recurrent thyroid cancer cases. This study further adds to the definition of the indication range of molecularly targeted therapies in precision oncology, which evolved from an increasing knowledge on cell signaling mechanisms involved in different types of cancer. Moreover, it highlights the use of broad genomics assessment for developing a personalized approach.

## Figures and Tables

**Figure 1 cancers-11-01185-f001:**
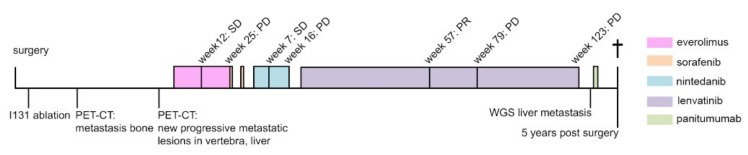
Timeline covering the final five-year-disease course of the present patient case. Having undergone standard therapy including surgery and radioactive iodine ablation, the patient developed progressive, metastatic, radioactive iodine refractory disease. An overview is depicted of administered targeted agents with evaluation of lesion response according to RECIST 1.1, first everolimus with SD at 12 weeks of use, PD 25 weeks, sorafenib for two weeks, nintedanib with SD at week 7, PD week 16, lenvatinib with PR at week 57, PD at week 79 and also week 123, and panitumumab for two weeks. WGS on metastatic material was done right before the last treatment regimen with panitumumab. PD, progressive disease. PR, partial response. SD, stable disease. WGS, whole genome sequencing.

**Figure 2 cancers-11-01185-f002:**
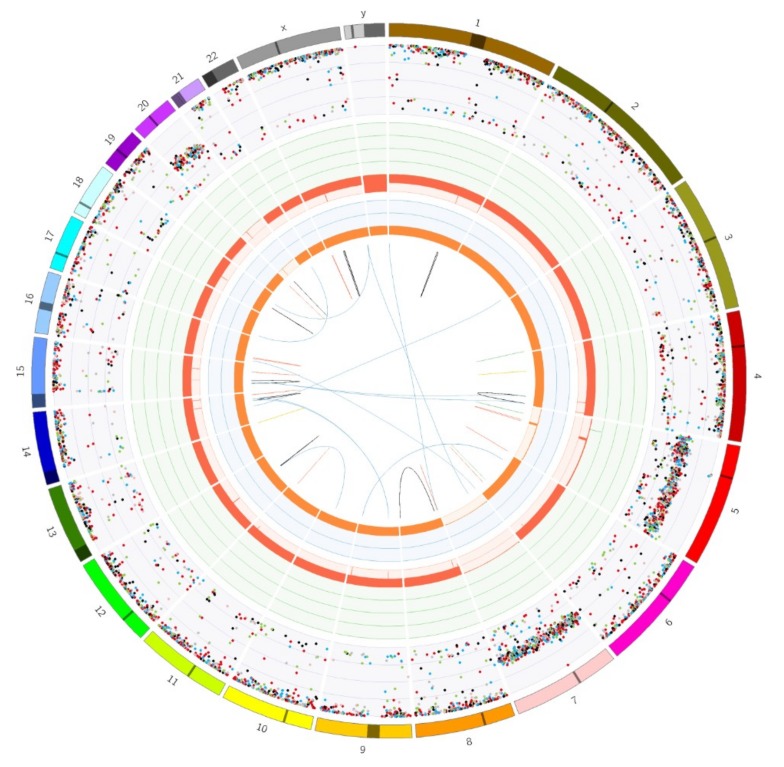
Circos plot showing a near-homozygous genome in HCC was detected by whole genome sequencing performed on biopsy material obtained from a metastatic hepatic lesion in the presented male patient case. LOH in all autosomes except for 5, 7, and 20. The outer first circle shows all chromosomes and the second circle shows all somatic variants (allele frequencies have been corrected for tumor purity and scale from 0% to 100%). The fourth circle represents the observed ‘minor allele copy numbers’ across the chromosome. The range of the chart is from 0 to 3 and the expected normal minor allele copy number is 1. Anything below 1 is shown as a loss (depicted in orange) and represents an LOH event.

**Figure 3 cancers-11-01185-f003:**
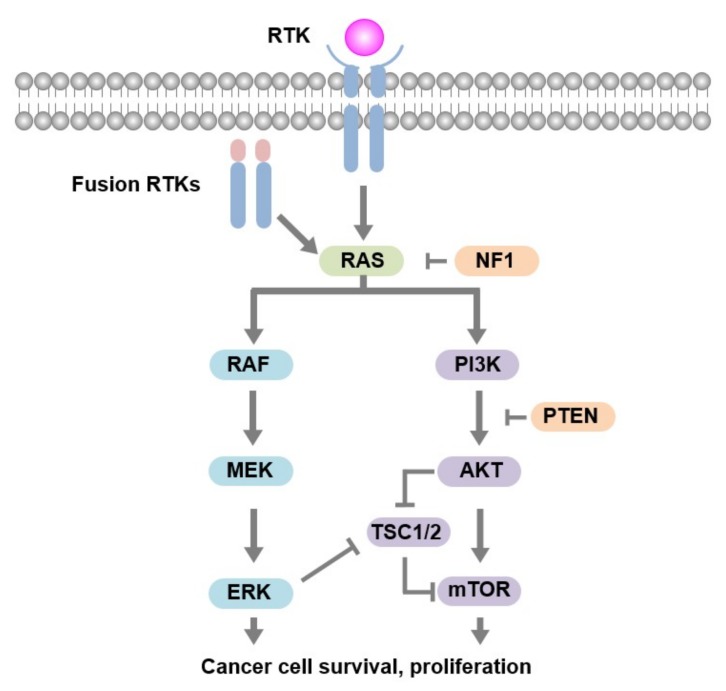
A simplified overview of the MAPK and PI3K pathway. Activation of pathways can occur by growth factors activating the RTKs or overexpression of RTKs. Mutations in the signaling pathway, in tumor suppressor genes, and fusion RTKs, can cause constitutive activation. RTK, receptor tyrosine kinase. MAPK, mitogen-activated protein kinase. PI3K, phosphatidylinositide 3-kinase. NF1, neurofibromin 1. PTEN, phosphatase and tensin homolog. TSC1/2, tuberous sclerosis complex 1/2. mTOR, mammalian target of rapamycin. VEGFR, vascular endothelial growth factor receptor. PDGFR-α, -β, platelet-derived growth factor receptor-α, -β, FGFR-1-3.

**Figure 4 cancers-11-01185-f004:**
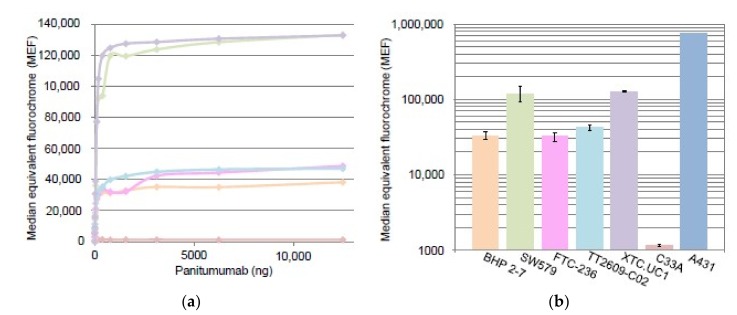
EGFR quantification of thyroid cancer cell lines by flow cytometry. (**a**) Represented by the purple line XTC.UC1 and the green line SW579. These show a relatively higher MEF as compared to the other thyroid cancer cell lines: TT2609-C02 (aqua line), FTC-236 (pink line), and BHP2-7 (orange line). Negative control: C33A (light red line). (**b**) Corresponding bar chart, including EGFR overexpressing A431 cells (dark blue bar) as a control.

**Figure 5 cancers-11-01185-f005:**
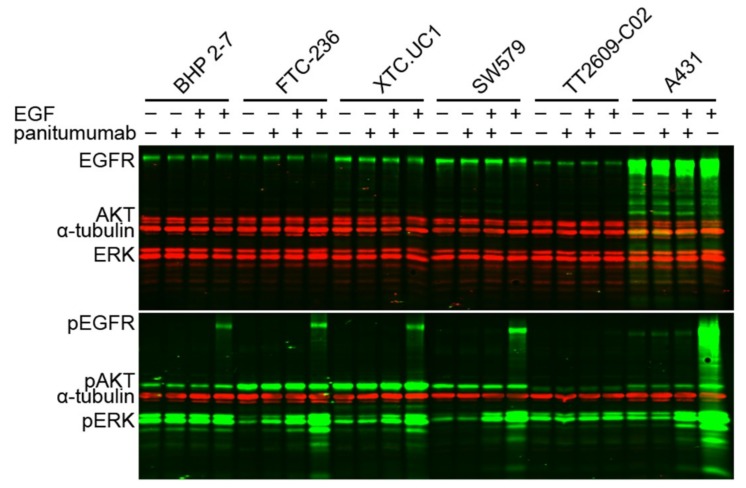
Immunofluorescent Western blots. Thyroid cancer cell lines (BHP 2–7, FTC-236, XTC.UC1, SW579, TT2609-C02) and the EGFR overexpressing control A431, were either serum-starved (−) or treated (+) with panitumumab for 1 h (20 µg/mL) and stimulated with (+) EGF from mouse for 10 minutes (100 ng/mL) or EGF unstimulated (−). A housekeeping protein, anti-α-tubulin antibody was used as loading control. The presence of EGFR, AKT, and ERK proteins in all cells, was confirmed by Western blotting. The same lysates were analyzed by Western blotting with phospho-EGFR (pEGFR), phospho-AKT (pAKT), and phospho-ERK (pERK) antibody. EGF induced EGFR phosphorylation (lane four of each cell line, compared to lane one) in all cell lines, except for TT2609-C02 (*NRAS* mutation). Panitumumab alone did not alter the EGFR phosphorylation status (lane 2, control). EGF-induced pEGFR phosphorylation was inhibited by panitumumab (third lane compared to the fourth). pERK showed a partial effect to panitumumab, in BHP 2–7, and TT2609-C02 pERK seems constitutively activated. pAKT seems constitutively activated in all cells (at low levels in TT2609-C02), and at significantly higher levels in the NHG harboring FTC-236 and XTC.UC1 cells in particular. Four gels were used in the same run (*BHP 2-7, FTC-236, XTC.UC1* loaded onto one gel for the protein, comprising the 12 vertical lanes on the upper left, and another one for phosphoprotein markers respectively, depicted on the left below. *SW579, TT2609-C02,* and *A431* were depicted onto the other two gels on the right). All gels were prepared simultaneously from the same mixture and all experimental procedures were run and processed simultaneously (also see Appendix A).

**Table 1 cancers-11-01185-t001:** Cancer cell line characteristics used for Western blot analysis.

Cell Line	Sex	Age	Origin	Localization	NHG	Variants	Ref.
BHP 2–7	f	uk	PTC	primary	N	*RET*/PTC1 rearrangement [61,62]	[63]
SW579	m	59	PDTC	primary	N	*TP53* c.827T > G, p. (Ile255Ser)	[64] ^1^
XTC.UC1	f	63	HCC	metas. in breast	Y [23]	*TP53* c.451C > A, NP_000537.3: p. (Pro151Thr) ^3^	[65] ^2^
FTC-236	m	42	FTC	neck LN metas.	Y [23]		[66]
TT2609-C02	m	57	FTC	primary	N	*NRAS* c.182A > G, p. (Gln61Arg) [67]	[68] ^1^
A431	f	85	EC	uk	uk	EGFR overexpression [69]	[70]

^1^ Obtained from DSMZ GmbH, Braunschweig, Germany. ^2^ Passage number p122 kindly provided by Dr. O.H. Clark, UCSF. ^3^ Likely, the pathogenic variant detected in this study (VAF > 80%). EC, epidermoid carcinoma. f, female. FTC, follicular thyroid carcinoma. HCC, Hürthle cell carcinoma. LN, lymph node. m, male. metas., metastasis. NHG, near-homozygous genome (Yes or No). PDTC, poorly differentiated thyroid carcinoma. PTC, papillary thyroid carcinoma. Ref., references. SSC, squamous cell carcinoma. uk, unknown.

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
