# Peer review of "Targeted Treatment Options of Recurrent Radioactive Iodine Refractory Hürthle Cell Cancer"

_cancers, 2019, doi:10.3390/cancers11081185_

Round 1

Reviewer 1 Report

The revision appears appropriate. The manuscript may now be published.

Reviewer 2 Report

I read the revised manuscript. I think that their responces in the point-by-point responce are satisfactory. Now for me the manuscript is suitable for the publication in Cancers Journal. Thank you

This manuscript is a resubmission of an earlier submission. The following is a list of the peer review reports and author responses from that submission.

Round 1

Reviewer 1 Report

In this report, Aydemirli and colleagues describe the pharmacological treatment of a patient affected by recurrent thyroid Hurtle cell carcinoma (HCC) in the context of a basket trial with off-label targeted therapies. The authors analyzed the clinical effects of the various drugs, including the presence/evolution of primary and metastatic lesions, collateral effects, and the time-to progression. In this patient, various TKI were used, including panitumumab, that showed no therapeutic efficacy. On bioptic material obtained fom a liver metastasis, the authors performed a WGS and found, according with previous reports, a near-homozigous genome (NHG), and TSC1 and NF1 mutations in homozigosity. These mutations were also found the in the primary tumor, with the exception of the TSC1 variant.

The authors went on to analyze a panel of human thyroid carcinoma cell lines, including an HCC cell line (XTC.UC1), to assess the effects of panitumumab, an EGFR monoclonal antibody. Similarly to the patient biopsy, this cell line is wt for RAS and expresses EGFR. Panitumumab inhibited EGFR phosphorylation induced by EGF, but ERK phosphorylation was only marginally affected. Moreover, AKT appears constitutively phosphorylated and insensitive to panitumumab treatment in this cell line.

The authors conclude that these data could represent the rationale against the use of panitumumab in HCC patients.

This report includes several interesting observations. However, the results  could be expanded and conclusions must be substantiated by further experimental data.

First, the unique HCC cell line used in this study has not been characterized by WGS, and it would be interesting to assess whether it carries other genetic/epigenetic alterations that are typical of HCC-in particular, genetic aterations that could explain the high constitutive pAKT levels-and to what extent it resembles the tumoral cells of the patient analyzed in this study. This would render this cell line a more suitable model for preclinical studies. Moreover, patient biopsies of primary and metatstatic tumor (if available) could be analyzed by IHC for pathway activation with anti-phospho-ERK and -AKT antibodies.

Second, the analysis of the effects of panitumumab on TC cell lines, including XTC.UC1, should include also biological responses, such as  cell proliferation, survival, migration/invasion, tumorigenicity in immunocompromised mice.

Third, based upon the activated signaling pathways, it would be important to assess the response of TC cell lines to various drugs used in the basket trial in which the patient analysed in this study was included. Of particular interest will be the XTC.UC1 response to PI3K/AKT/mTOR inhibitors, since this cell line displays constitutive activation of this signaling pathway.

Reviewer 2 Report

In the article entitled “Targeted Treatment Options of Recurrent Radioactive Iodine Refractory Hürthle Cell Cancer“ by Dr. Aydemirli and colleagues the authors describe the fate of a cancer patient who suffered on recurrent HCC and deceased after 5 years of treatment by a series of well established methods and drugs.

From a retrospective point of view on this case the authors try to explain why the various attempts to treat the disease were not successful.  They came to the conclusion: „This study further adds to define the indication range of molecularly targeted therapies in precision  oncology, evolving from an increasing knowledge on cell signaling mechanisms involved in different types of cancer. Moreover, it highlights the use of broad genomics assessment for developing a personalized approach.“

The conclusion is interesting. However, the manuscript needs some substantiation.

From  my point of view the article is written in a very confusing way. In the  Abstract, the aims and results of the study need to be more clearly explained. In the Introduction, the relevance of the unavailingly treated patient for this study needs further explanation.

Please  describe the information provided by Fig.2 more clearly and explain how  this information influenced the subsequent treating of the patient in detail. Obviously the absence of RAS variants      was the sole reason for applying panitumumab. Why is the whole Fig. 2  shown.

Whole  genome sequencing (WGS) was performed on a biopsy from a liver metastasis of the patient.  At the time of  taking the biopsy additional sites of tumor growth were present in the patient. Why did the WGS on only the biopsy challenge the application of panitumumab.

It is a nice idea to use cell lines in order to test whether a drug, which is selected because of a result of a WGS on a biopsy, may be applied to cure a patient.   Please show the most important common features of the biopsy and the cell lines used and explain the reasons for selecting the mentioned cell lines  in detail.

In  2.2, the rationale for non-responsiveness to panitumumab is demonstrated. A reasonable speculation about the application of a more useful drug in the patient’s situation may add considerable value to this article, if it is based on the result of the two WGS experiments. Attempts to justify the speculation would surely confirm the last sentence of the conclusion (lines 409/410).  

Reviewer 3 Report

In this study, the authors examined the clinical effects of different targeted treatment options in one patient with recurrent HCC. They performed whole genome sequencing on metastatic tumor material from liver at an advanced stage of the disease, finding a near-homozygous genome. Since the patient was wild type for RAS genes, they treated the patient with panitumumab, an inhibitor of EGFR. However, the disease progressed.

Then the authors analyzed the effects of panitumumab treatment on ERK and AKT signaling in NHG and non-NHG HCC cell line models. Authors should state more clearly the connection between the two parts through the text.

Overall the paper should be written with more logical connections and some specific points need to be addressed by the authors:

1. Introduction, page 2, lines 72-77, the authors should explain better why they chose to treat cancer thyroid cell lines with panitumumab and analyze ERK and AKT pathways.

2. Results, page 3-4, authors performed whole genome sequencing on a biopsy from a liver metastasis. However, no information about WGS results are reported (e.g. mean coverage, percentage of mutated reads, coverage of mutated genes).

Moreover, they found homozygous variants in TSC1 and in NF1 genes. Have these variants been validated? Have these variants already been reported in public databases and in literature?

By performing an NGS panel assay, the authors found that the mutation in NF1 gene was present also in the primary tumor, however, they still should provide more results for the panel and clarify if the mutation is a somatic mutation. In fact, it would be interesting to know if this mutation is a germline mutation.

3. Results, pages 6-7, western blotting analysis. The western blot analysis is highly descriptive and requires quantification. Western blot analysis of phosphorylated and total proteins should be performed on the same lanes of the same gel. Furthermore, quantification should be performed on replicates derived from different cell lysates.

4. In the conclusions authors stated that on the bases of the results on cell lines “recurrent HCC do not seem likely to benefit from panitumumab treatment”, however the connection is not clear and should be discussed better. They showed a constitutive activation of AKT pathway in tumor thyroid cell lines harboring a near homozygous genome. Do the authors have a hypothesis to explain the correlation between the acquisition of a near homozygous genome and the activation of AKT pathway?
